# A Nonlinear Finite-Time Robust Differential Game Guidance Law

**DOI:** 10.3390/s22176650

**Published:** 2022-09-02

**Authors:** Axing Xi, Yuanli Cai

**Affiliations:** School of Automation Science and Engineering, Xi’an Jiaotong University, Xi’an 710049, China

**Keywords:** guidance law, neuro-dynamic programming, differential game, nonlinear system

## Abstract

In this paper, a robust differential game guidance law is proposed for the nonlinear zero-sum system with unknown dynamics and external disturbances. First, the continuous-time nonlinear zero-sum differential game problem is transformed into solving the nonlinear Hamilton–Jacobi–Isaacs equation, a time-varying cost function is developed to reflect the fixed terminal time, and the robust guidance law is developed to compensate for the external disturbance. Then, a novel neural network identifier is designed to approximate the unknown nonlinear dynamics with online weight tuning. Subsequently, an online critic neural network approximator is presented to estimate the cost function, and time-varying activation functions are considered to deal with the fixed final time problem. An adaptive weight tuning law is given, where two additional terms are added to ensure the stability of the closed-loop nonlinear system and so as to meet the terminal cost at a fixed final time. Furthermore, the uniform ultimate boundedness of the closed-loop system and the critic neural network weights estimation error are proven based upon the Lyapunov approach. Finally, some simulation results are presented to demonstrate the effectiveness of the proposed robust differential game guidance law for nonlinear interception.

## 1. Introduction

In the modern battlefield, when the missile executes the interception mission, the fuel consumption, rapid attitude maneuver, and other uncertain factors may lead to the nonlinear dynamic characteristics of the system. Moreover, external disturbances, which may cause the system to lose control, are important factors in practice applications. Furthermore, the missile intercepts maneuvering targets with a fixed terminal time, which is a more effective attack. Therefore, it is of great significance to study the finite-time robust differential game law in nonlinear circumstances with external disturbance.

Traditional interception guidance strategies include proportional navigation guidance (PNG) [1] and augmented proportional navigation guidance (APNG) [2,3]. In these works, the interception problem was simplified into a two-dimensional plane and the target aircrafts were assumed to be non-maneuvering targets. With the development of the modern control theory, optimal guidance laws (OGLs), sliding mode control (SMC) guidance laws, and linear quadratic differential game (LQDG) guidance laws are investigated to solve the missile interception problem. In [4], first-order interceptor dynamics were taken into account, and an optimal guidance law (OGL) was proposed to control the impact time and impact angle. In [5], a combination of a line-of-sight (LOS) rate shaping technique and a second-order SMC was proposed. However, the target position was assumed to be known in advance. In [6], three-party pursuit and evasion guidance strategies were derived from a linear dynamic system, analytical solutions were derived via the Game Algebraic Riccati equation (GARE) and LQDG guidance laws were proposed. However, it is almost impossible, or even impossible, to acquire the analytic solution, due to the inherent nonlinearity. To circumvent the inherent nonlinearity of the system, in [7], an SDRE guidance law was proposed by solving the nonlinear Hamilton–Jacobi–Isaacs (HJI) equation. However, the external disturbance was neglected and it was difficult for the analytical solution to find its Nash equilibrium. Recently, an artificial neural network [8] has been applied to solve the nonlinear HJI equation for an unknown system. However, the finite-horizon optimal guidance law still remains an unsolved problem.

To solve the GARE, adaptive dynamic programming (ADP) techniques serve as powerful tools to solve the optimal control problems. In [9], a novel online mode-free reinforcement learning algorithm was proposed to solve the multiplayer non-zero-sum games, and the algebraic Riccati equation (ARE) was solved by an iterative algorithm. However, external disturbances were not considered. Unfortunately, in practice applications, external disturbances always exist. In [10], the author proposed a data-driven value iteration (VI) algorithm to handle the adaptive CT linear optimal output regulation. The optimal feedback control gain was learned by an online value iteration algorithm. However, the iterative algorithm requires a significant number of iterations within a fixed sampling interval to guarantee the stability of the system. In the above references, no matter what the mode-free integral RL algorithm or the data-driven value iteration algorithm, all the proposed methods were based on a solvable ARE. However, in most practice applications, external disturbances and parameter uncertainties always exist, which the ARE cannot obtain, in addition to the optimal controller.

For other contemporary approaches related to ADP, one can refer to refs. [11,12]. However, these are not applicable to practice applications with unknown or inaccurate system dynamics. The neuro-dynamic programming (NDP) technique is a typical method used to solve the nonlinear Hamilton–Jacobi–Isaacs (HJI) equation. In [13], the policy or value iteration-based NDP scheme was proposed to obtain the finite-horizon ε-optimal control for the discrete-time nonlinear system by using offline neural network (NN) training. In [14], an online algorithm based on policy iteration was proposed to attain the synchronous optimal policy with infinite horizon time for nonlinear systems with known dynamics. However, inadequate iterations within a sampling interval may lead to instability. To avoid this problem, in [15], a time-based NDP method was studied, and iteration-based optimal solutions were replaced by using the previous history of system states and cost function approximations. However, this NDP scheme was not suitable for finite-time nonlinear control; furthermore, external disturbances were not considered and only infinite-time optimal control was studied. In [16], an online concurrent learning NDP algorithm was presented to solve the two-player zero-sum game of nonlinear CT systems with unknown system dynamics and three NN approximators were tuned to learn about the value function corresponding to the optimal control strategies. However, two or more NNs led to an increase in computational complexity.

Although the above research has solved the HJB problem to a certain extent, unfortunately, there are few kinds of literature involved in the missile–target interception problem. In [17], a novel sliding mode adaptive neural network guidance law was proposed to intercept highly maneuvering targets. Aimed at the external disturbance caused by the target maneuvering, the RBF neural network was adopted to eliminate estimation errors without prior information about the target. Similarly, in [18], an adaptive NN-based scheme with an estimation cost function was proposed for solving the interception problem of the spacecraft with limited communication and external disturbances, but it did consider the target maneuverability. However, in these studies, only the missile interception strategy is considered, and the target control strategy is ignored.

In this paper, the nonlinear system of missile–target engagement with external disturbances is considered, and a time-varying cost function is designed for satisfying the terminal interception time. The uncertain nonlinear two-player zero-sum game is developed via the HJI equation based on differential game theory. The main contributions of this paper include two aspects. First, for solving the external disturbances problem, unlike the work [14], an extended robust interception guidance strategy for the missile to intercept the maneuvering target within a fixed final time is proposed. Second, two novel NNs are designed, one online NN identifier is designed to approximate the unknown nonlinear system, and the other critic NN is developed to approximate the cost function without policy or value iterations, while online learning is adopted. Finally, the nonlinear finite-time robust differential game guidance law is proposed.

The advantages of the proposed method of this paper are listed as follows:(1)The nonlinear guidance law for missile–target engagement within a fixed interception time is studied by using the differential game theory based on neuro-dynamic programming (NDP). More importantly, a time-varying cost function is reconstructed by a critic NDP with two additional terms added to ensure the stability of the nonlinear system and to meet the fixed interception time, which implies that the missile can intercept the target at different terminal times.(2)In practical applications, there are always external disturbances, and a robust interception guidance law is proposed to deal with this problem. Furthermore, inspired by the work [19], our proposed method extends the controller by considering the target.(3)Unlike the discrete system, our proposed method is the CT. Moreover, compared with the existing work [15,16], a clear advantage of our method is that a simpler critic NN structure is designed; thus, the computational burden is alleviated.

The remainder of this paper is organized as follows. The statement of the normal guidance problem is presented in part II. The robust control strategy of the nonlinear system with external disturbances is developed in part III. In part IV, an online NN identifier and a novel NDP-based approximator are presented. The stability of the nonlinear system is proved in part V. The nonlinear model of the missile–target engagement is established and numerical experiments are carried out to evaluate the performance of the proposed robust differential game guidance strategy in part VI. Part VII presents some conclusions.

## 2. Problem Formulation

In this paper, the two players are the missile and the target, which are described in detail in part VI. For the finite-time nonlinear two-player zero-sum differential game, the object of the missile input $\bar{u}\left(t\right)$ is to minimize the cost function, while the target input $\bar{w}\left(t\right)$ is to maximize the cost function in a specified time. The continuous-time (CT) uncertain nonlinear two-player zero-sum differential game is now presented as
(1)$$\dot{x}\left(t\right)=f\left(x\left(t\right)\right)+g\left(x\left(t\right)\right)\left(\bar{u}\left(t\right)+\bar{d}\left(x\right)\right)+k\left(x\left(t\right)\right)\left(\bar{w}\left(t\right)+\bar{p}\left(x\right)\right)$$
where $x\left(t\right)\in R^{n}$, $\bar{u}\left(t\right)\in R^{m}$, and $\bar{w}\left(t\right)\in R^{q}$ represent the system state vector, the control input of the missile and the target, respectively. $f\left(x\right)\in R^{n}$ denotes the internal system dynamics, $g\left(x\right)\in R^{n\times m}\;,\;\mathrm{and}\;k\left(x\right)\in R^{n\times q}$ are the control coefficient matrices, with f(x),g(x) and k(x) are locally Lipschitz, respectively. $\bar{d}\left(x\right)\;\mathrm{and}\;\bar{p}\left(x\right)$ represent external disturbances, which are both bounded by known functions $d_{M}\left(x\right)$ and $p_{M}\left(x\right)$, i.e., $\left\|\bar{d}\left(x\right)\right\|\leq d_{M}\left(x\right)$ and $\left\|\bar{p}\left(x\right)\right\|\leq p_{M}\left(x\right)$. Furthermore, we assume that $d\left(x\right)=R_{1}^{\frac{1}{2}}\bar{d}\left(x\right)$ and $p\left(x\right)=\gamma \bar{d}\left(x\right)$, with $R_{1}$ and γ are symmetric positive definite matrices, respectively.

The nominal system (without external disturbances) of the system (1) can be described as
(2)$$\dot{x}\left(t\right)=f\left(x\left(t\right)\right)+g\left(x\left(t\right)\right)u\left(t\right)+k\left(x\left(t\right)\right)w\left(t\right)$$

We assume that f+gu+kw is Lipschitz continuous on a set Ω and the system (2) is controllable.

Considering external disturbances in the nonlinear system (1), for the nominal system (2), the finite-time two-player zero-sum differential game cost function is defined as
(3)$$\begin{array}{c} V\left(x,t_{0}\right)=\varphi \left(x\left(t_{f}\right),t_{f}\right)+\int_{t_{0}}^{t_{f}}r\left(x\left(t\right),u\left(t\right),w\left(t\right)\right)dt\\ V\left(x,t_{f}\right)=\varphi \left(x\left(t_{f}\right),t_{f}\right) \end{array}$$
where $r\left(x\left(t\right),u\left(t\right),w\left(t\right)\right)=Q\left(x\right)+u^{T}R_{1}u-\frac{1}{2\gamma^{2}}w^{T}w$, with $Q\left(x\right)=d_{M}^{2}\left(x\right)+R_{2}p_{M}^{2}\left(x\right)+x^{T}Q_{1}x$; $Q_{1}$ is a semi-positive function. The terminal cost, external disturbances, control efforts of the missile and the target, the system state, and fixed terminal time are chosen as the performance evaluation indicators. $\varphi \left(x\left(t_{f}\right),t_{f}\right)$ reflects the terminal cost between the missile with the target. The term Q(x) reflects external disturbances and the system state simultaneously, which is positively defined. Moreover, $R_{2}$ represents the influence of the target disturbance. $R_{1}$ reflects the missile control effort. γ reflects the target control effort. All parameters Q(x), $R_{1},$ $R_{2},$ and γ are positively defined. The goal of this paper is to find the saddle point of the cost function (3).

**Remark** **1.**
*First, unlike the infinite-time scenario, the terminal cost*

$V\left(x,t_{0}\right)$

*is a time-varying function.*

$\varphi \left(x\left(t_{f}\right),t_{f}\right)$

*is needed to guarantee the finite-time scenario. Second, external disturbances are considered in the cost function via adopting a positive constant*

$R_{2}$

*and the robust control problem can be addressed by designing the finite-time guidance strategy of the nominal system (2).*


Assuming that $V\left(x,t\right)\in C^{1}$, an infinitesimal equivalent to (3) can be derived as
(4)$$-\frac{\partial V\left(x,t\right)}{\partial t}=r\left(x,u,d\right)+\frac{\partial V^{T}\left(x,t\right)}{\partial x}\left(f\left(x\right)+g\left(x\right)u+k\left(x\right)w\right)$$
when $t_{0}=t_{f}$, the terminal cost function can be expressed as
(5)$$V\left(x,t_{f}\right)=\Psi \left(x\left(t_{f}\right),t_{f}\right)$$

The Hamiltonian function of the nonlinear system (2) can be defined as
(6)$$H\left(x,u,v\right)=V_{t}+r\left(x,u,v\right)+V_{x}^{T}\left(f\left(x\right)+g\left(x\right)u\left(t\right)+k\left(x\right)w\left(t\right)\right)$$
where $V_{t}=\frac{\partial V\left(x,t\right)}{\partial t}$ and $V_{x}=\frac{\partial V\left(x,t\right)}{\partial x}$. It can be clearly observed that the Hamiltonian function includes a time-dependent term $V_{t}$.

In the Nash equilibrium theory, the saddle point with respect to the optimal control pair (u(t),w(t)) can be obtained by
(7)$$H\left(x,u^{*},w\right)\leq H\left(x,u^{*},w^{*}\right)\leq H\left(x,u,w^{*}\right)$$

If one recalls the classical optimal game theory, both optimal controllers can be solved by using stationary conditions $\frac{\partial H\left(\cdot \right)}{\partial u}=0$ and $\frac{\partial H\left(\cdot \right)}{\partial w}=0$, which yields
(8)$$\{\begin{array}{c} u^{*}\left(x\right)=-\frac{1}{2}R_{1}^{-1}g^{T}\left(x\right)V_{x}^{*}\\ w^{*}\left(x\right)=\gamma^{2}k^{T}\left(x\right)\;V_{x}^{*} \end{array}$$
where $V^{*}\left(x,t\right)$ is the optimal two-player zero-sum game cost function, which is the saddle point of the cost function, such that
(9)$$\begin{array}{c} V^{*}\left(x,t\right)=\underset{w\left(t\right)}{\max}\underset{u\left(t\right)}{\min}V\left(x,t\right)\\ V^{*}\left(x,t_{f}\right)=\varphi \left(x\left(t_{f}\right),t_{f}\right) \end{array}$$

By substituting optimal strategy (8) into Equation (4), the Hamilton–Jacobi–Isaacs (HJI) equation reduces to
(10)$$\begin{array}{c} V_{t}^{*}+Q\left(x\right)+V_{x}^{*}{}^{T}f\left(x\right)-\frac{1}{4}V_{x}^{*}{}^{T}g\left(x\right)R_{1}^{-1}g^{T}\left(x\right)V_{x}^{*}\\ +\frac{\gamma^{2}}{2}V_{x}^{*}{}^{T}k\left(x\right)k^{T}\left(x\right)V_{x}^{*}=0 \end{array}$$

**Remark** **2.***For the linear system case, the HJB equation can be easily solved by the Riccati equation* [19]*. However, it is difficult or even impossible to attain the mathematical solution of the HJI Equation (10), when system dynamics exist in nonlinear terms. Moreover, the fixed final time*
$t_{f}$
*is provided in this nonlinear system for solving the inadequate iterations problem, a novel time-based online optimal guidance law design is proposed and the system dynamics are demonstrated.*

## 3. The Nonlinear Finite-Time Robust Differential Game Guidance Law

In this part, the nonlinear finite-time robust differential game guidance law is presented. First, for coping with the robust stabilization problem of the system (1) with external disturbances, a robust controller is designed. Then, the finite-time NDP-based optimal guidance strategy is designed.

### 3.1. Robust Controller Design of Uncertain Nonlinear Differentia Games

By extending the work [20], two feedback gains $\pi_{1}\;\mathrm{and}\frac{1}{2\pi_{2}}$ are added to the optimal feedback control (8) of the system (2) for the missile and the target, respectively. The robust optimal feedback control yields as follows:(11)$$\{\begin{array}{c} {\bar{u}}^{*}\left(x\right)=\pi_{1}u^{*}\left(x\right)=-\frac{1}{2}\pi_{1}R_{1}^{-1}g^{T}\left(x\right)V_{x}^{*}\\ \;{\bar{w}}^{*}\left(x\right)=\frac{1}{2\pi_{2}}w^{*}\left(x\right)=\frac{1}{2\pi_{2}}\gamma^{2}k^{T}\left(x\right)\;V_{x}^{*} \end{array}$$

Here, some lemmas are presented for indicating that the robust optimal control has an infinite gain margin.

**Lemma** **1.***For the nominal system (2), the optimal control strategy given by (11) can ensure that the closed-loop system is asymptotically stable for*$\pi_{1}\geq \frac{1}{2}$*and*$\pi_{2}\geq 1$.

**Proof** **of** **Lemma** **1.**The optimal cost function $V^{*}\left(x,t\right)=J^{*}\left(x,t\right)$ is selected as the Lyapunov function. In light of (3), it is easily found that $V^{*}\left(x,t\right)\;$ is positive definite. By combining (10) and (11), the derivative of the $V^{*}\left(x,t\right)$ along the trajectory of the closed-loop system yields
(12)$${\dot{J}}^{*}\left(x,t\right)=V_{t}^{*}+V_{x}^{*}{}^{T}\left(f\left(x\right)+g\left(x\right)\bar{u}\left(t\right)+k\left(x\right)\bar{w}\left(t\right)\right)\;=V_{t}^{*}-Q\left(x\right)-V_{t}^{*}+\frac{1}{4}V_{x}^{*}{}^{T}g\left(x\right)R_{1}^{-1}g^{T}\left(x\right)V_{x}^{*}\;-\frac{\gamma^{2}}{2}V_{x}^{*}{}^{T}k\left(x\right)k^{T}\left(x\right)V_{x}^{*}-\frac{1}{2}\pi_{1}g\left(x\right)R_{1}^{-1}g^{T}\left(x\right)V_{x}^{*}+\frac{1}{2\pi_{2}}\gamma^{2}k\left(x\right)k^{T}\left(x\right)\;V_{x}^{*}\;\begin{array}{c} \;=-Q\left(x\right)-\frac{1}{2}\left(\pi_{1}-\frac{1}{2}\right){\left\|R_{1}^{-\frac{1}{2}}g^{T}\left(x\right)V_{x}^{*}\right\|}^{2}-\frac{1}{2}\left(1-\frac{1}{\pi_{2}}\right){\left\|\gamma k^{T}\left(x\right)V_{x}^{*}\right\|}^{2} \end{array}$$

Hence, whenever $\pi_{1}\geq \frac{1}{2}$, $\pi_{2}\geq 1$ and x≠0, ${\dot{J}}^{*}\left(x,t\right)<0$. □

**Theorem** **1.**
*For the system (1), there exists two positive gains,*

$\pi_{1}^{*}\geq 1$

*and*

$\pi_{2}^{*}\geq \frac{2R_{2}}{2R_{2}+1}$

*, with*

$R_{2}>1$

*, such that for any*

$\pi_{1}>\pi_{1}^{*}$

*and*

$\pi_{2}>\pi_{2}^{*}$

*, the robust control (11) ensures that the closed-loop system (1) is asymptotically stable.*


**Proof** **of** **Theorem** **1.**The optimal cost function $L\left(x,t\right)=V^{*}\left(x,t\right)$ is selected as the Lyapunov function, and the derivative of the $V^{*}\left(x,t\right)$ along the trajectory of the closed-loop system can be obtained as
(13)$$L\left(x,t\right)=V_{t}^{*}+V_{x}^{*}{}^{T}\left(f\left(x\right)+g\left(x\right)\left(\bar{u}\left(t\right)+\bar{d}\left(x\right)\right)+k\left(x\right)\left(\bar{w}\left(t\right)+\bar{p}\left(x\right)\right)\right)$$

Based on (12), (13) can be rewritten as
(14)$$\dot{L}\left(x,t\right)=-Q\left(x\right)-\frac{1}{2}\left(\pi_{1}-\frac{1}{2}\right){\left\|V_{x}^{*}{}^{T}g\left(x\right)R_{1}^{-\frac{1}{2}}\right\|}^{2}-\frac{1}{2}\left(1-\frac{1}{\pi_{2}}\right){\left\|V_{x}^{*}{}^{T}g\left(x\right)\gamma \right\|}^{2}+V_{x}^{*}{}^{T}g\left(x\right)\bar{d}\left(x\right)+V_{x}^{*}{}^{T}k\left(x\right)\bar{p}\left(x\right)\;\leq -x^{T}Q_{1}x-\left(\frac{1}{2}\left(\pi_{1}-\frac{1}{2}\right){\left\|V_{x}^{*}{}^{T}g\left(x\right)R_{1}^{-\frac{1}{2}}\right\|}^{2}+\frac{1}{2}\left(1-\frac{1}{\pi_{2}}\right){\left\|V_{x}^{*}{}^{T}g\left(x\right)\gamma \right\|}^{2}-\left\|V_{x}^{*}{}^{T}g\left(x\right)R_{1}^{-\frac{1}{2}}\right\|d_{M}\left(x\right)\;-\left\|V_{x}^{*}{}^{T}k\left(x\right)\gamma \right\|d_{M}\left(x\right)+d_{M}^{2}\left(x\right)+R_{2}d_{M}^{2}\left(x\right)\right)$$

If $z={\left[d_{M}\left(x\right),\;p_{M}\left(x\right),\left\|V_{x}^{*}{}^{T}g\left(x\right)R_{1}^{-\frac{1}{2}}\right\|,\;\left\|V_{x}^{*}{}^{T}k\left(x\right)\gamma \right\|\right]}^{T}$; thus, (14) can be rewritten as
(15)$$\dot{L}\left(x,t\right)\leq -x^{T}Q_{1}x-z^{T}\Theta z$$
where $\Theta =\left[\begin{array}{cccc} 1 & 0 & -\frac{1}{2} & 0\\ 0 & R_{2} & 0 & -\frac{1}{2}\\ -\frac{1}{2} & 0 & \frac{1}{2}\left(\pi_{1}-\frac{1}{2}\right) & 0\\ 0 & -\frac{1}{2} & 0 & \frac{1}{2}\left(1-\frac{1}{\pi_{2}}\right) \end{array}\right]$.

According to the Lyapunov function, the determinant Θ≥0 represents the $\dot{L}\left(x,t\right)\leq 0$ and implies that the closed-loop system is asymptotically stable. Thus, it can be concluded that $\pi_{1}^{*}\geq 1$ and $\pi_{2}^{*}\geq \frac{2R_{2}}{2R_{2}+1}$ can ensure the positive definiteness of Θ. When $\pi_{1}>\pi_{1}^{*}$ and $\pi_{2}>\pi_{2}^{*}$, the closed-loop system is asymptotically stable. □

### 3.2. Finite-Time NDP-Based Optimal Guidance Strategy

First, a novel online NN identifier is proposed to approximate the unknown system dynamics. Next, a critical NDP-based approximator is utilized to estimate the cost function within a fixed final time and an online adaptive weight tuning law is proposed with additional terms to guarantee the stability of the nonlinear system. Finally, combining the identified system and estimated cost function, the finite-horizon optimal differential guidance strategy is derived.

#### 3.2.1. NN Identifier

System dynamics are necessary for developing guidance laws of the nonlinear two-player zero-sum differential games. However, system dynamics may be unknown in practice applications. To overcome this problem, a novel online NN identifier is designed. Based on the NN universal function approximation property, the nonlinear system can be represented as
(16)$$f\left(x\right)=W_{f}^{T}\sigma_{f\left(x\right)}+\varepsilon_{f}\;g\left(x\right)=W_{g}^{T}\sigma_{g\left(x\right)}+\varepsilon_{g}\;k\left(x\right)=W_{k}^{T}\sigma_{k\left(x\right)}+\varepsilon_{k}$$
where $W_{f}\in R^{f\times N},W_{g}\in R^{g\times N}$, and $W_{k}\in R^{k\times N}$ are ideal weight matrices. $\sigma_{f\left(x\right)}\in R^{N},\;\sigma_{g\left(x\right)}\in R^{N}$, and $\sigma_{k\left(x\right)}\in R^{N}$ denote NN activation function vectors, *N* is the number of hidden layer neurons, and $\varepsilon_{f},\;\varepsilon_{g},$ and $\varepsilon_{k}$ represent NN approximation errors.

Then, the nominal system (2) can be represented by using (16) as
(17)$$\dot{x}=f\left(x\right)+g\left(x\right)u+k\left(x\right)w={\left[\begin{array}{c} W_{f}\\ W_{g}\\ W_{k} \end{array}\right]}^{T}\left[\begin{array}{ccc} \sigma_{f\left(x\right)} & 0 & 0\\ 0 & \sigma_{g\left(x\right)} & 0\\ 0 & 0 & \sigma_{k\left(x\right)} \end{array}\right]\left[\begin{array}{c} 1\\ u\\ w \end{array}\right]+\varepsilon_{f}+\varepsilon_{g}u+\varepsilon_{k}w=W_{I}^{T}\sigma_{I\left(x\right)}\bar{\xi }+\varepsilon_{I}$$

Because the ideal NN weights are typically unknown, we define the state estimator as follows:(18)$$\dot{\hat{x}}={\hat{W}}_{I}^{T}\sigma_{I\left(x\right)}\bar{\xi }+K\tilde{x}$$
where ${\hat{W}}_{I}$ represents the estimate of the $W_{I}$. $\tilde{x}=x-\hat{x}\in R^{N}$ denotes the state estimation error. K is a design parameter, which can guarantee the stability of the NN identifier.

From (17) and (18), the derivative of the state estimation error yields
(19)$$\dot{\tilde{x}}=\dot{x}-\dot{\hat{x}}={{\tilde{W}}_{I}}^{T}\sigma_{I\left(x\right)}\bar{\xi }+\varepsilon_{I}-K\tilde{x}$$

Inspired by [9], in order to make the approximated NN identifier weight matrix close to its ideal value, the online tuning law is given by
(20)$${\dot{\hat{W}}}_{I}\left(t\right)=-\alpha_{1}{\hat{W}}_{I}\left(t\right)+\sigma_{I\left(x\right)}\bar{\xi }\tilde{x}$$
where $\alpha_{1}$ is the learning rate of the critical NN.

Next, by defining ${\tilde{W}}_{I}=W_{I}-{\hat{W}}_{I}$, the identifier weight estimation error yields the following equation by using (20):(21)$$\dot{{\tilde{W}}_{I}}\left(t\right)=\alpha_{1}{\hat{W}}_{I}\left(t\right)-\sigma_{I\left(x\right)}\bar{\xi }\tilde{x}$$

**Theorem** **2.**
*For the NN identifier (18), let the initial ideal critic NN weight*

$W_{I}$

*reside in a compact set by selecting the proposed NN weight tuning law provided (20). There exists a positive tuning parameter (*

$\alpha_{1}>0$

*). Then, the identification (19) and the weight estimation errors*

${\tilde{W}}_{I}\left(t\right)$

*are uniformly ultimately bounded (UUB) with a fixed terminal time.*


**Proof** **of** **Theorem** **2.**Let the Lyapunov function candidate be as follows:
(22)$$J_{I}=\frac{1}{2}{\tilde{x}}^{T}\tilde{x}+\frac{1}{2}tr\left({{\tilde{W}}_{I}}^{T}{\tilde{W}}_{I}\right)$$

Then,
(23)$$\dot{J_{I}}={\tilde{x}}^{T}\dot{\tilde{x}}+tr\left({{\tilde{W}}_{I}}^{T}\dot{{\tilde{W}}_{I}}\right)\leq \frac{1}{2}{\left\|\tilde{x}\right\|}^{2}+\frac{1}{2}{\left\|{\tilde{\varepsilon }}_{I}\right\|}^{2}-\lambda_{min}\left(K\right){\left\|\tilde{x}\right\|}^{2}+\alpha_{1}tr\left({{\tilde{W}}_{I}}^{T}{\tilde{W}}_{I}\right)-\alpha_{1}tr\left({{\tilde{W}}_{I}}^{T}{\tilde{W}}_{I}\right)\;\leq -\left(\lambda_{min}\left(K\right)-\frac{1}{2}\right){\left\|\tilde{x}\right\|}^{2}-\frac{\alpha_{1}}{2}{\left\|{\tilde{W}}_{I}\right\|}^{2}+\varepsilon_{IM}$$
where $\varepsilon_{IM}=\frac{\alpha_{1}}{2}W_{I}^{2}+\frac{1}{2}{\left\|\varepsilon_{I}\right\|}^{2}$, the eigenvalue *K* and $\alpha_{1}$ are the design parameters that guarantee the stability of the system. Therefore, when $\lambda_{min}\left(K\right)\geq \frac{1}{2},$ and $\alpha_{1}\geq 0$, the following inequalities hold
(24)$$\left\|\tilde{x}\right\|>\sqrt{\frac{\varepsilon_{IM}}{\left(\lambda_{min}\left(K\right)-\frac{1}{2}\right)}}\;\mathrm{or}\;\left\|{\tilde{W}}_{I}\right\|>\sqrt{\frac{2\varepsilon_{IM}}{\alpha_{1}}}$$

It can be observed from (24) that $\left\|\tilde{x}\right\|$ bound can be decreased by increasing the eigenvalue *K*. Therefore, $\left\|\tilde{x}\right\|$ is quantified by selecting the minimum eigenvalue $\lambda_{min}\left(K\right)$. According to (19) and the relationship between $\left\|\tilde{x}\right\|$ and $\left\|{\tilde{W}}_{I}\right\|$, the smaller $\left\|\tilde{x}\right\|$ will enforce the $\left\|{\tilde{W}}_{I}\right\|$ to converge into a small bound. Therefore, we have $\dot{J_{I}}\leq 0$ and it can be concluded that $\left\|\tilde{x}\right\|$ and $\left\|{\tilde{W}}_{I}\right\|$ are UUB.

This completes the proof. □

#### 3.2.2. NDP-Based Guidance Strategy

In order to confront the nonlinear HJI function, according to the universal approximation property of the neural network, the optimal cost function can be reconstructed by a critic NDP on a compact set as
(25)$$V\left(x,t\right)=W_{V}^{T}h\left(x,t_{f}-t\right)+\varepsilon_{V}\left(x,t\right)$$
where $W_{V}\in R^{V\times N}\;is$ the ideal weight matrix. $h\left(x,t_{f}-t\right)\in R^{N}$ denotes NN activation function vectors, *N* is the number of hidden layer neurons, and $\varepsilon_{v}$ represents the NN approximation error.

Thus, the terminal cost function can be expressed as
(26)$$V\left(x,t_{f}\right)=W_{V}^{T}h\left(x\left(t_{f}\right),0\right)+\varepsilon_{V}\left(x,t_{f}\right)$$

**Remark** **3.**
*The critic NN activation function*

h(x,t)

*and its gradient*

∇h(x,t)

*are upper bounded, i.e.,*

$\|h\left(x,t\right)\|\leq h_{M}$

*and*

$\|\nabla h\left(x,t\right)\|\leq h_{dM}$

*, with*

$h_{M}$

*and*

$h_{dM}$

*positive constants. The critic NN weight W is upper bounded, i.e.,*

$\|W_{V}\|\leq W_{VM}$

*, with*

$W_{VM}$

*being a positive constant. The critic NN approximation error*

$\varepsilon_{V}\left(x\right)$

*and its gradient*

$\nabla \varepsilon_{V}\left(x\right)$

*are upper bounded, i.e.,*

$\|\varepsilon_{V}\left(x\right)\|\leq \varepsilon_{VM}$

*and*

$\|\nabla \varepsilon_{V}\left(x\right)\|\leq \varepsilon_{dVM}$

*, with*

$\varepsilon_{VM}$

*and*

$\varepsilon_{dVM}$

*positive constants. Obviously, the activation function is a time-varying function, which can maintain the fixed final time.*


Next, the partial derivation of V(x,t) with respect to *x* and *t* can be obtained, respectively.
(27)$$\begin{array}{c} V_{x}=\nabla h_{x}^{T}\left(x,t_{f}-t\right)W_{V}+\nabla_{x}\varepsilon_{V}\left(x,t\right)\\ V_{t}=\nabla h_{t}^{T}\left(x,t_{f}-t\right)W_{V}+\nabla_{t}\varepsilon_{V}\left(x,t\right) \end{array}$$
where $\nabla h_{x}^{T}\left(x,t_{f}-t\right)=\partial h\left(x,t_{f}-t\right)/\partial x$, $\nabla h_{t}^{T}\left(x,t_{f}-t\right)=\partial h\left(x,t_{f}-t\right)/\partial t$, $\nabla_{x}\varepsilon_{V}\left(x,t\right)=\partial \varepsilon_{V}\left(x,t_{f}-t\right)/\partial x$, and $\nabla_{t}\varepsilon_{V}\left(x,t\right)=\partial \varepsilon_{V}\left(x,t_{f}-t\right)/\partial t$.

Therefore, by substituting (27) into (8), we then obtain the differential game guidance strategy as
(28)$$\{\begin{array}{c} u^{*}\left(x\right)=-\frac{1}{2}R_{1}^{-1}g^{T}\left(x\right)V_{x}^{*}=-\frac{1}{2}R_{1}^{-1}g^{T}\left(x\right)\left(\nabla h_{x}^{T}\left(x,t_{f}-t\right)W_{V}+\nabla_{x}\varepsilon_{V}\left(x,t\right)\right)\\ w^{*}\left(x\right)=\gamma^{2}k^{T}\left(x\right)\;V_{x}^{*}=\gamma^{2}k^{T}\left(x\right)\left(\nabla h_{x}^{T}\left(x,t_{f}-t\right)W_{V}+\nabla_{x}\varepsilon_{V}\left(x,t\right)\right) \end{array}$$

By substituting (28) into (10), the HJI function can be rewritten as
(29)$$H\left(x,u,w\right)=W_{V}^{T}\nabla h_{t}^{T}\left(x,t_{f}-t\right)+Q\left(x\right)+W_{V}^{T}\nabla h_{x}\left(x,t_{f}-t\right)f\left(x\right)\;\begin{array}{c} -\frac{1}{4}W_{V}^{T}\nabla h_{x}\left(x,t_{f}-t\right)D_{1}\left(x\right)\nabla h_{x}^{T}\left(x,t_{f}-t\right)W_{V}+\frac{\gamma^{2}}{2}W_{V}^{T}\nabla h_{x}\left(x,t_{f}-t\right)k\left(x\right)k^{T}\left(x\right)\nabla h_{x}^{T}\left(x,t_{f}-t\right)W_{V}+\varepsilon_{HJB}\left(x,t\right) \end{array}$$
where $D_{1}\left(x\right)=g\left(x\right)R_{1}^{-1}g^{T}\left(x\right)$ and
(30)$$\varepsilon_{HJB}\left(x,t\right)=\nabla_{t}\varepsilon_{V}\left(x,t\right)+\frac{1}{2}W_{V}^{T}h\left(x,t_{f}-t\right)\nabla_{x}\varepsilon_{V}\left(x,t\right)D_{1}\left(x\right)\nabla_{x}\varepsilon_{V}\left(x,t\right)-\gamma^{2}W_{V}^{T}h\left(x,t_{f}-t\right)\nabla_{x}\varepsilon_{V}\left(x,t\right)D_{1}\left(x\right)\nabla_{x}\varepsilon_{V}\left(x,t\right)\;-\frac{1}{2}\nabla_{x}\varepsilon_{V}^{T}\left(x,t\right)D_{1}\left(x\right)\nabla h_{x}^{T}\left(x,t_{f}-t\right)W_{V}-\frac{1}{4}\nabla_{x}\varepsilon_{V}^{T}\left(x,t\right)D_{1}\left(x\right)\nabla_{x}\varepsilon_{V}\left(x,t\right)\;+\gamma^{2}\nabla_{x}\varepsilon_{V}^{T}\left(x,t\right)k\left(x\right)k^{T}\left(x\right)\nabla h_{x}^{T}\left(x,t_{f}-t\right)W_{V}+\frac{\gamma^{2}}{2}\nabla_{x}\varepsilon_{V}^{T}\left(x,t\right)k\left(x\right)k^{T}\left(x\right)\nabla_{x}\varepsilon_{V}\left(x,t\right)\;+\frac{1}{4}W_{V}^{T}\nabla h_{x}\left(x,t_{f}-t\right)D_{1}\left(x\right)\nabla_{x}\varepsilon_{V}\left(x,t\right)\;+\nabla_{x}\varepsilon_{V}^{T}\left(x,t\right)f\left(x\right)-\frac{\gamma^{2}}{2}W_{V}^{T}\nabla h_{x}\left(x,t_{f}-t\right)k\left(x\right)k^{T}\left(x\right)\nabla_{x}\varepsilon_{V}\left(x,t\right)$$

Because ideal NN weights are typically unknown, we define the estimated cost function $\hat{V}\left(x,t\right)$ as follows
(31)$$\hat{V}\left(x,t\right)={\hat{W}}_{V}^{T}h\left(x,t_{f}-t\right)$$

The estimated terminal cost function is
(32)$$\hat{V}\left(x,t_{f}\right)={\hat{W}}_{V}^{T}h\left(\hat{x}\left(t_{f}\right),0\right)$$
where ${\hat{W}}_{V}$ denotes the estimate of the $W_{V}$ and $h\left(\hat{x}\left(t_{f}\right),0\right)$ is the activation function with the estimated terminal state $\hat{x}\left(t_{f}\right)$.

Next, the partial derivation of the estimated cost function $\hat{V}\left(x,t\right)$ with respect to *x* and *t* can be obtained, respectively.
(33)$$\begin{array}{c} {\hat{V}}_{x}=\nabla h_{x}^{T}\left(x,t_{f}-t\right){\hat{W}}_{V}\\ {\hat{V}}_{t}=\nabla h_{t}^{T}\left(x,t_{f}-t\right){\hat{W}}_{V} \end{array}$$
where ${\hat{V}}_{t}=\frac{\partial V\left(x,t\right)}{\partial t}$ and ${\hat{V}}_{x}=\frac{\partial V\left(x,t\right)}{\partial x}$.

Then, by applying (33) to (8), the estimated differential game guidance strategy can be rewritten as
(34)$$\{\begin{array}{c} \hat{u}\left(x\right)=-\frac{1}{2}R_{1}^{-1}g^{T}\left(x\right){\hat{V}}_{x}=-\frac{1}{2}R_{1}^{-1}g^{T}\left(x\right)(\nabla h_{x}^{T}\left(x,t_{f}-t\right){\hat{W}}_{V}\\ \hat{w}\left(x\right)=\gamma^{2}k^{T}\left(x\right){\hat{V}}_{x}=\gamma^{2}k^{T}\left(x\right)\nabla h_{x}^{T}\left(x,t_{f}-t\right){\hat{W}}_{V} \end{array}$$

By applying (34) to (10), the estimated HJB function yields
(35)$$\hat{H}\left(x,u,w\right)={\hat{W}}_{V}^{T}\nabla h_{t}^{T}\left(x,t_{f}-t\right)+{\hat{W}}_{V}^{T}\nabla h_{x}\left(x,t_{f}-t\right)\hat{f}\left(x\right)\;\begin{array}{c} -\frac{1}{4}{\hat{W}}_{V}^{T}\nabla h_{x}\left(x,t_{f}-t\right){\hat{D}}_{1}\left(x\right)\nabla h_{x}^{T}\left(x,t_{f}-t\right){\hat{W}}_{V}+\frac{\gamma^{2}}{2}{\hat{W}}_{V}^{T}\nabla h_{x}\left(x,t_{f}-t\right)\hat{k}\left(x\right){\hat{k}}^{T}\left(x\right)\nabla h_{x}^{T}\left(x,t_{f}-t\right){\hat{W}}_{V}=e_{c} \end{array}$$

In order to obtain the optimal differential game guidance strategy, we define the estimated terminal cost error as
(36)$$e_{tf}=\Psi \left(x\left(t_{f}\right),0\right)-{\hat{W}}_{V}^{T}h\left(\hat{x}\left(t_{f}\right),0\right)=W_{V}^{T}\tilde{h}\left(x\left(t_{f}\right),0\right)+{\tilde{W}}_{V}^{T}h\left(\hat{x}\left(t_{f}\right),0\right)+\varepsilon_{t_{f}}$$
where $\tilde{h}\left(x\left(t_{f}\right),0\right)=h\left(x\left(t_{f}\right),0\right)-h\left(\hat{x}\left(t_{f}\right),0\right)$.

Moreover, in order to guarantee the estimated NN weight ${\hat{W}}_{V}$, which is approximated to the ideal NN weight $W_{V}$, and after combining the time-varying nature of the cost function and the estimated terminal cost error, the total NN approximation error is defined as
(37)$$E_{total}=\frac{1}{2}e_{c}^{T}e_{c}+\frac{1}{2}e_{t_{f}}^{4}$$

By using the gradient descent algorithm, a novel simplified weight tuning law is proposed with additional terms to ensure the stability of the nonlinear system (1), as follows:(38)$${\dot{\hat{W}}}_{V}=-\alpha_{2}\frac{\left(\hat{\omega }+\frac{1}{2}sng\left(\hat{\omega }\right)\right)}{{\left(1+{\hat{\omega }}^{T}\hat{\omega }\right)}^{2}}e_{c}+\alpha_{3}\frac{\left(\hat{\xi }+\frac{1}{2}sng\left(\hat{\xi }\right)\right)}{{\left(1+{\hat{\xi }}^{T}\hat{\xi }\right)}^{2}}e_{t_{f}}^{3}\;\begin{array}{c} +\frac{\alpha_{4}}{2}\nabla h_{x}^{T}\left(x,t_{f}-t\right)\left({\hat{D}}_{1}\left(x\right)-2\gamma^{2}{\hat{k}}^{T}\left(x\right)k^{T}\left(x\right)\right)Q\left(x,t\right) \end{array}$$
where $\hat{\omega }=\nabla h_{t}^{T}\left(x,t_{f}-t\right)+\nabla h_{x}\left(x,t_{f}-t\right)\hat{f}\left(x\right)+\frac{1}{2}\nabla h_{x}\left(x,t_{f}-t\right)\left({\hat{D}}_{1}\left(x\right)+{\hat{k}}^{T}\left(x\right)k^{T}\left(x\right)\right)\nabla h_{x}^{T}\left(x,t_{f}-t\right){\hat{W}}_{V}$, $\hat{\xi }=h\left(\hat{x}\left(t_{f}\right),t_{f}\right)$

**Remark** **4.**
*It is important to mention that the first term in (38) is used to minimize the squared residual error. The second term is used to minimize the terminal cost estimation error. The last term is used to guarantee that the system states remain bounded.*


## 4. Stability Analysis

In order to prove the stability of the weight tuning law and the nonlinear system by choosing the optimal strategy, and without loss of generality, the weight estimation error of the critic NN is defined as ${\tilde{W}}_{V}=W_{V}-{\hat{W}}_{V}$; then, we have ${\dot{\tilde{W}}}_{V}=-{\dot{\hat{W}}}_{V}$. Therefore, the approximated HJI is
(39)$$\hat{H}\left(x,u,w\right)=-{\tilde{W}}_{V}^{T}\nabla h_{t}^{T}\left(x,t_{f}-t\right)-{\tilde{W}}_{V}^{T}\nabla h_{x}\left(x,t_{f}-t\right)\hat{f}\left(x\right)-W_{V}^{T}\nabla h_{x}\left(x,t_{f}-t\right)\tilde{f}\left(x\right)\;+\frac{1}{2}{\tilde{W}}_{V}^{T}\nabla h_{x}\left(x,t_{f}-t\right)\left({\hat{D}}_{1}\left(x\right)-2\gamma^{2}{\hat{k}}^{T}\left(x\right)k^{T}\left(x\right)\right)\nabla h_{x}^{T}\left(x,t_{f}-t\right){\hat{W}}_{V}\;\begin{array}{c} \;+\frac{1}{4}W_{V}^{T}\nabla h_{x}\left(x,t_{f}-t\right)\left(\tilde{D}\left(x\right)-2\gamma^{2}{\tilde{k}}^{T}\left(x\right)\tilde{k}\left(x\right)-4\gamma^{2}{\tilde{k}}^{T}\left(x\right)\hat{k}\left(x\right)\right)\nabla h_{x}^{T}\left(x,t_{f}-t\right)W_{V}+\varepsilon_{HJB} \end{array}$$

**Assumption** **1.**
*For the nonlinear system (1) with the cost function (2) and the optimal guidance law (34), let the value function*

V(x,t)

*be the Lyapunov function and continuously differentiable, when*

(40)
$$\dot{V}\left(x,t\right)=V_{x}\left(x,t\right)\dot{x}+V_{t}\left(x,t\right)+V_{x}\left(x,t\right)\left(f\left(x\right)+g\left(x\right)u^{*}+k\left(x\right)w^{*}\right)<0$$


*Exists and it is clear that the inequation holds*

(41)
$$V_{t}\left(x,t\right)+V_{x}\left(x,t\right)\left(f\left(x\right)+g\left(x\right)u^{*}+k\left(x\right)w^{*}\right)<-Q\left(x,t\right)$$



**Theorem** **3.**
*For the nonlinear system (1) with the ideal HJI Equation (9), let the updated law for the NN-based identifier and NDP-based cost function approximator be provided by (20) and (38), respectively and the estimated optimal guidance laws are given in (34). The existence of the positive constants*

$B_{Q_{x}}$

*,*

$B_{W_{V}}$

*,*

$B_{\tilde{x}}$

*and*

$B_{W_{I}}$

*mean that the identification error, weight estimation error and the controller are UUB.*


**Proof** **of** **Theorem** **3.**Select the Lyapunov candidate function as
(42)$$J_{V}=J_{aV}+J_{bV}+J_{cV}+J_{dV}$$
where $J_{aV}=\frac{\alpha_{2}}{4}{\left({\tilde{x}}^{T}\Lambda \tilde{x}\right)}^{2}+\frac{\alpha_{3}}{4}{\left(tr\left({{\tilde{W}}_{I}}^{T}\Lambda {\tilde{W}}_{I}\right)\right)}^{2}$, $J_{bV}=\frac{1}{2}{\tilde{W}}_{V}^{T}\Pi {\tilde{W}}_{V}$, $J_{cV}=\alpha_{4}V\left(x,t\right)$, $J_{dV}=\frac{1}{2}{\tilde{x}}^{T}\Xi \tilde{x}+\frac{1}{2}tr\left({{\tilde{W}}_{I}}^{T}\Lambda {\tilde{W}}_{I}\right)$.

It can be observed that $J_{V}>0$.

First, the derivation of $J_{aV}$ with respect to time is given by
(43)$${\dot{J}}_{aV}\leq {\left\|\Lambda \right\|}^{2}\alpha_{2}\left({\tilde{x}}^{T}\tilde{x}\right){\tilde{x}}^{T}\dot{\tilde{x}}+{\left\|\Lambda \right\|}^{2}\alpha_{3}tr\left({{\tilde{W}}_{I}}^{T}{\tilde{W}}_{I}\right)tr\left({{\tilde{W}}_{I}}^{T}\dot{{\tilde{W}}_{I}}\right)\;\leq -\alpha_{2}\left(\lambda_{min}\left(K_{I}\right)-\frac{3}{2}-\frac{\alpha_{3}}{4\alpha_{2}}\right){\left\|\Lambda \right\|}^{2}{\left\|\tilde{x}\right\|}^{4}+\frac{1}{4}{\left\|\Lambda \varepsilon_{I}\right\|}^{2}{\left\|\tilde{u}\right\|}^{2}\;-\frac{\alpha_{3}}{2}\left(\frac{\alpha_{I}}{2}-1-\frac{\alpha_{2}+\alpha_{3}}{4\alpha_{3}}{\left\|\zeta_{I}\right\|}^{4}\right){\left\|\Lambda \right\|}^{2}{\left\|{\tilde{W}}_{I}\right\|}^{4}+\varepsilon_{cM}$$
where $\alpha_{2}>0$, $\alpha_{3}>0$ and $\alpha_{I}>4+\frac{\alpha_{2}+\alpha_{3}}{4\alpha_{3}}\|\zeta_{I}\|{}^{4}$, $K_{I}$ satisfies $\lambda_{min}\left(K_{I}\right)>\frac{3}{2}+\frac{\alpha_{3}}{4\alpha_{2}},$ $\varepsilon_{cM}=\frac{\alpha_{3}\alpha_{I}}{4}{\left\|\Lambda \right\|}^{2}W_{I,M}^{4}+\frac{{\left\|\Lambda \right\|}^{2}\alpha_{2}\left\|\varepsilon_{I}\right\|{}^{2}}{4}$.

The derivation of $J_{bV}$ is governed by
(44)$${\dot{J}}_{bV}={\tilde{W}}_{V}^{T}\Pi {\dot{\tilde{W}}}_{V}\;\leq \left\|\Pi \right\|\left({\tilde{W}}_{V}^{T}\alpha_{2}\frac{\hat{\omega }}{{\left(1+{\hat{\omega }}^{T}\hat{\omega }\right)}^{2}}e_{c}-\alpha_{3}{\tilde{W}}_{V}^{T}\frac{\hat{\xi }}{{\left(1+{\hat{\xi }}^{T}\hat{\xi }\right)}^{2}}e_{t_{f}}-\frac{\alpha_{4}}{2}{\tilde{W}}_{V}^{T}\nabla h_{x}^{T}\left(x,t_{f}-t\right)\left({\hat{D}}_{1}\left(x\right)+\hat{k}\left(x\right)k^{T}\left(x\right)\right)Q\left(x,t\right)\right)\;\leq -\frac{\alpha_{2}\|\Pi \|}{8}\frac{{\tilde{W}}_{V}^{T}\hat{\omega }{\hat{\omega }}^{T}{\tilde{W}}_{V}}{{\left(1+{\hat{\omega }}^{T}\hat{\omega }\right)}^{2}}+\frac{\alpha_{2}W_{VM}^{4}\lambda_{max}^{2}\left(R^{-1}\right)\nabla h_{xM}^{4}\sigma_{I,M}^{4}}{{\left(1+{\hat{\omega }}^{T}\hat{\omega }\right)}^{2}}{\left\|{\tilde{W}}_{I}\right\|}^{4}\;-\frac{\alpha_{4}\|\Pi \|}{2}{\tilde{W}}_{V}^{T}\nabla h_{x}^{T}\left(x,t_{f}-t\right)\left({\hat{D}}_{1}\left(x\right)-2\gamma^{2}\hat{k}\left(x\right)k^{T}\left(x\right)\right)Q\left(x,t\right)+\varepsilon_{VH}$$
where
$$0<\alpha_{2}\leq \frac{3\alpha_{2}{\left(1+{\hat{\omega }}^{T}\hat{\omega }\right)}^{2}\left(\lambda_{min}\left(\hat{\xi }{\hat{\xi }}^{T}\right)+\frac{1}{2}\right)}{\lambda_{max}^{2}\left(\nabla h_{x}\left(x,t_{f}-t\right)\right)\left({\hat{D}}_{1}\left(x\right)-2\gamma^{2}\hat{k}\left(x\right)k^{T}\left(x\right)\right)\nabla h_{x}^{T}\left(x,t_{f}-t\right){\left(1+{\hat{\xi }}^{T}\hat{\xi }\right)}^{2}}$$ and $\varepsilon_{VH}=\frac{\alpha_{2}\|\Pi \|}{2}\frac{\varepsilon_{HJB}^{2}}{{\left(1+{\hat{\omega }}^{T}\hat{\omega }\right)}^{2}}+3{\left(W_{V}^{T}\tilde{h}\left(x\left(t_{f}\right),0\right){\tilde{h}}^{T}\left(x\left(t_{f}\right),0\right)W_{V}+\varepsilon_{V}^{2}\left(x,t_{f}\right)\right)}^{2}+\frac{15}{2}{\left(\alpha_{3}\frac{\varepsilon_{V}^{2}\left(x,t_{f}\right)}{{\left(1+{\hat{\xi }}^{T}\hat{\xi }\right)}^{2}}+\frac{W_{V}^{T}\tilde{h}\left(x\left(t_{f}\right),0\right){\tilde{h}}^{T}\left(x\left(t_{f}\right),0\right)W_{V}}{{\left(1+{\hat{\xi }}^{T}\hat{\xi }\right)}^{2}}\right)}^{2}+\frac{\alpha_{2}W_{VM}^{4}\lambda_{max}^{4}\left(R^{-1}\right)h_{xM}^{4}\varepsilon_{IH}^{4}}{2\left({\left(1+{\hat{\omega }}^{T}\hat{\omega }\right)}^{2}\right)}$. $\lambda_{min}\left(R^{-1}\right)$ and $\lambda_{max}\left(R^{-1}\right)$ is the minimum and the maximum eigenvalue of the matrix $R^{-1}$.

Next, we have
(45)$${\dot{J}}_{cV}=\alpha_{4}\left(V_{t}\left(x,t\right)+V_{x}\left(x,t\right)\left(f\left(x\right)+g\left(x\right)\hat{u}+k\left(x\right)\hat{w}\right)\right)\;=\alpha_{4}(V_{t}\left(x,t\right)+V_{x}\left(x,t\right)\left(f\left(x\right)-\frac{1}{2}g\left(x\right)R^{-1}{\hat{g}}^{T}\nabla h_{x}\left(x,t_{f}-t\right){\hat{W}}_{V}+\gamma^{2}k\left(x\right){\hat{k}}^{T}\left(x\right)\nabla h_{x}\left(x,t_{f}-t\right){\hat{W}}_{V}\right)$$

Thus,
(46)$$\dot{J_{V}}={\dot{J}}_{aV}+{\dot{J}}_{bV}+{\dot{J}}_{cV}+{\dot{J}}_{dV}\;\leq \alpha_{4}\left(V_{t}\left(x,t\right)+V_{x}\left(x,t\right)\left(f\left(x\right)-\frac{1}{2}g\left(x\right)R^{-1}{\hat{g}}^{T}\left(x\right)\right)\nabla h_{x}\left(x,t_{f}-t\right){\hat{W}}_{V}+\gamma^{2}k\left(x\right){\hat{k}}^{T}\left(x\right)\nabla h_{x}\left(x,t_{f}-t\right){\hat{W}}_{V}\right)\;+\frac{\alpha_{2}W_{VM}^{4}\lambda_{max}^{4}\left(R^{-1}\right)\nabla h_{xM}^{4}\sigma_{I,M}^{4}}{16{\left(1+{\hat{\omega }}^{T}\hat{\omega }\right)}^{2}}\left\|\Lambda \right\|{\left\|{\tilde{W}}_{I}\right\|}^{4}-\frac{\alpha_{I}}{2}\left\|\Xi \right\|{\left\|{\tilde{W}}_{I}\right\|}^{2}+\left\|\Xi \right\|\varepsilon_{IM}-\frac{1}{2}(\alpha_{2}\frac{{\hat{\omega }}^{T}\hat{\omega }}{{\left(1+{\hat{\omega }}^{T}\hat{\omega }\right)}^{2}}\;+\alpha_{3}\frac{{\hat{\xi }}^{T}\hat{\xi }}{{\left(1+{\hat{\xi }}^{T}\hat{\xi }\right)}^{2}})\left\|\Pi \right\|\left\|{\tilde{W}}_{V}\right\|{}^{2}-\frac{\alpha_{4}}{2}{\tilde{W}}_{V}^{T}\nabla h_{x}^{T}\left(x,t_{f}-t\right)\left({\hat{D}}_{1}\left(x\right)-2\gamma^{2}\hat{k}\left(x\right)k^{T}\left(x\right)\right)Q\left(x,t\right)+\varepsilon_{VH}\;-\alpha_{2}\left(\lambda_{min}\left(K_{I}-\frac{3}{2}-\frac{\alpha_{3}\left\|\zeta_{I}\right\|{}^{4}}{8\alpha_{2}}\right){\left\|\Lambda \right\|}^{2}{\left\|\tilde{x}\right\|}^{4}+\varepsilon_{IM}\right)-\frac{\alpha_{3}}{4}\left(\alpha_{I}-\frac{3}{2}-\frac{\alpha_{2}}{\alpha_{3}}\left\|\zeta_{I}\right\|{}^{4}\right){\left\|\Lambda \right\|}^{2}\left\|{\tilde{W}}_{I}\right\|{}^{4}-(\lambda_{min}\left(K_{I}\right)-\frac{1}{2})\left\|\Xi \right\|{\left\|\tilde{x}\right\|}^{2}\;\leq -\frac{\alpha_{4}}{5}{\left\|Q\left(x,t\right)\right\|}^{2}-\frac{5\alpha_{4}\left(W_{VM}^{2}\nabla h_{xM}^{2}+1\right)W_{IM}^{2}\lambda_{max}^{2}\left(R^{-1}\right)\sigma_{I,M}^{4}}{16}\left\|{\tilde{W}}_{I}\right\|{}^{2}\;-\frac{\alpha_{2}}{4}\left\|\Lambda \right\|\frac{{\hat{\omega }}^{T}\hat{\omega }}{{\left(1+{\hat{\omega }}^{T}\hat{\omega }\right)}^{2}}\left\|{\tilde{W}}_{V}\right\|{}^{2}-\frac{\alpha_{2}}{8}\left\|\Lambda \right\|\frac{1}{{\left(1+{\hat{\omega }}^{T}\hat{\omega }\right)}^{2}}\left\|{\tilde{W}}_{V}\right\|{}^{2}+\varepsilon_{TC}$$
where $$0<\alpha_{4}\leq \min\left(\frac{5\lambda_{min}\left(1+{\hat{\omega }}^{T}\hat{\omega }\right)+\frac{5}{2}}{32\nabla h_{xM}^{6}Q_{min}W_{VM}^{4}\lambda_{max}^{2}\left(R^{-1}\right){\left(1+{\hat{\omega }}^{T}\hat{\omega }\right)}^{2}},1\right)$$
$$\Xi =\frac{5\alpha_{4}\left(W_{VM}^{2}\nabla h_{xM}^{2}+1\right)W_{IM}^{2}\sigma_{I,M}^{4}\lambda_{max}^{2}\left(R^{-1}\right)}{4\alpha_{I}}I$$
$$\Pi =\frac{5}{8\alpha_{2}W_{VM}^{4}\lambda_{max}^{2}\left(R^{-1}\right)\nabla h_{xM}^{4}}I\;\varepsilon_{TC}=\varepsilon_{WM}+\|\Xi \|\varepsilon_{IM}+\varepsilon_{cM}+\frac{5\alpha_{4}}{4}\|\varepsilon_{xM}\|{}^{2}$$

$\varepsilon_{IM}=\varepsilon_{fM}+\varepsilon_{gM}u_{M}^{*}+\varepsilon_{hM}d_{M}^{*}$. $u_{M}^{*}$ and $d_{M}^{*}$ are the upper bounds of the optimal guidance strategy $u^{*}$ and $d^{*}$.

Therefore, when the following conditions hold, the first derivation of $J_{V}$ is less than zero.
$$\left\|Q\left(x,t\right)\right\|>\sqrt{\frac{5\varepsilon_{TC}}{\alpha_{4}}}=B_{Q_{x}}\;\mathrm{or}\;\left\|{\tilde{W}}_{V}\right\|>\sqrt{\frac{2\varepsilon_{TC}}{\alpha_{4}\nabla h_{x}^{2}}}=B_{W_{V}}\;\mathrm{or}\;\left\|\tilde{x}\right\|>\sqrt{\frac{\varepsilon_{TC}}{\left(\lambda_{min}\left(K_{I}-\frac{1}{2}\right))\|\Xi \right\|}}=B_{\tilde{x}}\;\mathrm{or}\;\left\|{\tilde{W}}_{I}\right\|>\;\sqrt{\frac{16\varepsilon_{TC}}{5\alpha_{4}\left(W_{VM}^{2}\nabla h_{x}^{2}+1\right)W_{IM}^{2}\lambda_{max}^{2}\left(R^{-1}\right)\sigma_{I,M}^{2}}}=B_{W_{I}}$$

This completes the proof. □

**Remark** **5.***The eigenvalue K,*$\alpha_{1}$*,*$\alpha_{2}$*,* $\alpha_{3}$*, and*$\alpha_{4}$*are the tuning parameters for guaranteeing the lower bound of* $B_{Q_{x}}$*,*$B_{W_{V}}$*,* $B_{\tilde{x}}$*and*$B_{W_{I}}$*, which can quantify the bound of the system. In additions, from the proof, we can observe that the estimated optimal guidance laws given in (34) can ensure the system is UUB. Thus, by combining the robust optimal feedback control (11), the complete nonlinear finite-time robust differential game guidance law is provided.*

## 5. Application

A missile–target engagement scenario is considered in this section. The engagement geometry of the missile–target is shown in Figure 1, where the *X-Y* plane represents the Cartesian reference frame. The variables V and A denote the speed and the normal acceleration of the missile and the target, respectively. *α* and *β* denote the flight angles of the missile and the target, respectively. The variables *r* and *θ* represent the missile–target distance and the line of sight (*LOS*) angle, and *LOS* angular rate is $\dot{\theta }$ donated by σ. u and w are the control vectors perpendicular to the velocity of the missile, and the target.

The engagement occurs in the terminal guidance phase and all the participators are assumed to neglect the effect of gravity and have constant velocity. The nonlinear kinematics of the missile and the target, in a polar coordinate system, is given by
(47)$$V_{r}=\dot{r}=V_{T}cos\left(\beta -\theta \right)-V_{M}cos\left(\alpha -\theta \right)$$
(48)$$\sigma =\dot{\theta }=(V_{T}sin\left(\beta -\theta \right)-V_{M}sin\left(\alpha -\theta \right))/r$$
where $V_{r}$ represents the closing velocity.

The first-order dynamics of the missile are considered, and the motions of the missile are as follows:(49)$${\dot{x}}_{M}=V_{M}cos\alpha$$
(50)$${\dot{y}}_{M}=V_{M}sin\alpha$$
(51)$$\dot{\alpha }=\frac{a_{M}}{V_{M}}$$
(52)$${\dot{a}}_{M}=\frac{u_{M}-a_{M}}{\tau_{M}}$$
where $\left(x_{M},y_{M}\right)$ is the position of the missile along with the Cartesian reference frame. $a_{M}$ is the lateral acceleration of the missile and $\tau_{M}$ is a time constant.

Similarly, the motion equations of the target are as follows:(53)$${\dot{x}}_{T}=V_{T}cos\beta$$
(54)$${\dot{y}}_{T}=V_{T}sin\beta$$
(55)$$\dot{\beta }=\frac{a_{T}}{V_{T}}$$
(56)$${\dot{a}}_{T}=\frac{u_{T}-a_{T}}{\tau_{T}}$$
where $\left(x_{T},y_{T}\right)$ is the position of the target along with the Cartesian reference frame. $a_{M}$ is the lateral acceleration of the target and $\tau_{T}$ is a time constant.

To obtain the capture zone, the guidance principle is adopted as follows.

**Definition** **2.**
*Zero effort miss distance (ZEMD) is the closest distance between the missile and the target at an instant*

t

*, while the missile and the target do not impose any control and the agents continue to perform their scheduled maneuver strategy from the current time to the endgame. The ZEMD is computed as*

(57)
$$r_{miss}\left(t\right)=\frac{r^{2}\sigma }{\sqrt{V_{r}^{2}+r^{2}\sigma^{2}}}$$



In this scenario, the missile applies its optimal strategy (34) to minimize the ZEMD, while the target applies its optimal strategy (34) to maximize the ZEMD. In addition, from (57), we can observe that if σ tends to be zero, the ZEMD also tends to be zero. Meanwhile, the closing $V_{r}$ is also less than zero. Therefore, to ensure that the missile successfully intercepts the target, the following two conditions hold, which will be verified in simulations:(58)$$\sigma \to 0,\;V_{r}<0$$

Based on the two conditions mentioned above, by choosing the $x={\left[\theta \;\sigma \right]}^{T}$ as the system state, differentiating the Equation (48) with respect to time, the nonlinear system can be given by
(59)$$\{\begin{array}{c} \dot{x_{1}}=x_{2}\\ \dot{x_{2}}=-\frac{2V_{r}}{r}x_{2}-\frac{cos\left(\beta -\theta \right)}{r}u_{M}+\frac{cos\left(\alpha -\theta \right)}{r}w_{T} \end{array}$$

**Remark** **6.**
*From (59), it can be observed that when*

r

*is close to 0, the nonlinear terms*

f(x), g(x), and k(x)

*tend to be infinite. It will mean that the system is broken and the proposed differential game guidance laws do not work. Thus, a minimized ZEMD (*

$r_{min}$

*) should be designed, which means that when the displacement between the missile and the target is less or equal to the*

$r_{min}=0.5\;m$

*, the missile completes the interception mission. Moreover, in the practice application, the missile has an actual killing radius; thus, this design is reasonable.*


**Remark** **7.**
*From (59), it also can be found that when*

|(β−θ)|=π/2

*and*

|(α−θ)|=0

*, no matter what the proposed guidance laws change, the nonlinear system is an unstable equilibrium. Therefore, the domain where the differential game-based guidance laws are applicable is given by*

(60)
$$\varphi =\left\{x:\;\left|\left(\beta -\theta \right)\right|\neq \frac{\pi }{2},\left|\left(\alpha -\theta \right)\right|\neq 0,r\neq 0,V_{r}<0\right\}$$



## 6. Simulation Results

In this section, some experiments are designed to verify the proposed finite-time robust differential game guidance strategy. To further explain the performance of the proposed differential game guidance strategy based on NDP, some simulation experiments are designed and analyzed. The initial engagement occurs in the terminal guidance phase. The speeds of the missile and the target are *V_M_ =* 700 m/s, and *V_T_ =* 400 m/s, respectively. The initial position of the target is at (0 m, 0 m), and the position of the missile is (2500 m, 0 m). The initial flight path angles of the missile and the target are α=70° and β=150°, respectively. The time constants are $\tau_{M}=0.1\;\mathrm{and}\;\tau_{T}=0.1$, respectively. The weight parameters $R_{1}=5$, $R_{2}=50,\gamma =10\;\mathrm{and}\;t_{f}=7\;s$ are set. $Q_{x}=20{\left(x_{1}^{2}+x_{2}^{2}+\tau^{2}\right)}^{2}$, where $\tau =t_{f}-t$.

Furthermore, the initial parameters of the critic NN are selected inside $w_{V}\in \left[0\;1\right]$ randomly and the critic NN identifier activation function is designed as $\sigma_{I\left(x\right)}={\left[1\;x_{1}^{2}\;x_{1}x_{2}\;x_{2}^{2}\;x_{1}^{3}\;x_{1}^{2}x_{2}\;x_{1}x_{2}^{2}\;x_{2}^{3}\;x_{1}^{4}\;x_{1}^{3}x_{2}\;x_{1}^{2}x_{2}^{2}\;x_{2}\;x_{2}^{3}\;x_{2}^{4}\right]}^{T}$, which refers to [18,21]. In addition, polynomial basis functions can better approximate the nonlinear system. The initial parameters of the critic NN are $w_{V}={\left[10\;10\;10\;10\;10\;10\;10\;10\right]}^{T}$ and the critic NN vector activation function for estimating the cost function is designed as
$$h\left(x,t_{f}-t\right)={\left[x_{1}^{2}\exp\left(-\tau \right)\;x_{2}^{2}\exp\left(-\tau \right)\;x_{1}x_{2}\tau \;x_{1}^{4}\exp\left(-\tau \right)\;x_{2}^{2}\exp\left(-\tau \right)\;x_{1}^{3}x_{2}\;x_{1}^{2}x_{2}\;x_{1}x_{2}^{2}\right]}^{T}.$$

The learning rates are $\alpha_{1}=0.01,\;\alpha_{2}=0.05,\;\alpha_{3}=0.55\;\mathrm{and}\;,\;\alpha_{4}=0.15$, respectively. The experiment environments are performed on a PC platform with i7-9750H CPU, using Matlab 2020b.

### 6.1. Effect of the Proposed Finite-Time Differential Game Guidance Strategy without Unknown Uncertainties

In this case, the performance of the proposed finite-time differential game guidance strategy is verified without unknown uncertainties and the missile and the target choose the optimal guidance strategy (34). More importantly, the *ZEMD* and the lateral acceleration of the agents are the performance evaluation indicators and we have rewritten the physical meaning. The performance of the *ZEMD* tends to be zero, indicating that the missile intercepts the target successfully. The performance of the lateral accelerations of the agents is limited in the range ±100 g and stable change indicates that the agents can be reasonably controlled. The simulation results for this engagement scenario are shown in Figure 2, Figure 3 and Figure 4.

Figure 2a shows the trajectories of the missile and the target by selecting the proposed finite-time differential game guidance strategy (34). Figure 3b presents the change in the relative distance between the missile and the target and it can be observed that the *ZEMD* is less than 0.5 m. By bombining (a) and (b), the results reveal that the missile intercepts the target successfully.

Both the angular rate and the relative velocity are presented in Figure 3. From Figure 3a, it can be observed that the state variable (the angular rate) bound is zero at about 4.5 s, which leads the ZEMD to be zero according to the Equation (57) and Figure 2b. σ tends to be zero, which can ensure that the missile intercepts the target. Furthermore, the negative $V_{r}$ guarantees the relative distance between the missile and the target tends to be zero in Figure 3b. Moreover, sharp changes between σ and $V_{r}$ are reasonable due to the system characteristics. Furthermore, it also verifies Definition 2, which mentioned that $\sigma \to 0,\;\mathrm{and}\;V_{r}<0$.

Figure 4 presents lateral acceleration curves of the target and the missile. It can be observed that both lateral accelerations are maintained within a reasonable range, which can ensure that the missile intercepts the target. However, due to the system characteristics, the lateral accelerations decrease sharply at the end of the engagement.

The convergence curves of critic NN weights are presented in Figure 5 and it can be observed that the critic NN weights finally converge to stable coefficients. The result reveal that critic NN weights can guarantee the stability of the closed-loop nonlinear system.

### 6.2. Engagement without Unknown Uncertainties for a Maneuvering Target

To further verify the effectiveness of the proposed guidance strategy dealing with the target executing other forms of target maneuvers, the following experiment is presented. In this experiment, the target is expected to perform a sin-wave maneuver with a magnitude of 10 g and the missile still selects the guidance law (34). The simulation results for this engagement scenario are shown in Figure 6, Figure 7 and Figure 8.

Figure 6a shows the engagement trajectories of the missile confronting the target, which is expected to perform a sin-wave maneuver by selecting the proposed guidance law (34). Figure 6b presents the change in the relative distance between the missile and the target and it can be observed that the final ZEMD is less than 0.5 m. The results reveal that the missile intercepts the maneuvering target successfully.

Curves of the angular rate and the range rate are presented in Figure 7. The results reveal the same conclusion as case 1.

Figure 8 presents the lateral acceleration curves of the target and the missile. It can be observed that both lateral accelerations are maintained within a reasonable range, which can ensure that the missile intercepts the target. Furthermore, compared with Figure 4a, due to the maneuvering target, higher acceleration demands of the missile are needed.

### 6.3. Engagement with Unknown Uncertainties for a Maneuvering Target

To further verify how the proposed finite-time robust guidance strategy deals with unknown uncertainties, the following experiment is presented. In this experiment, the missile selects the robust differential game guidance law (11), and the target is expected to perform a sin-wave maneuver with a magnitude of 10 g. Furthermore, external disturbances, with a uniform distribution between −0.2 and 0.2, are considered in both input vectors. The simulation results for this engagement scenario are shown in Figure 9, Figure 10 and Figure 11.

Figure 9a shows the trajectories of the missile confronting the target, which is expected to perform a sin-ware maneuver by selecting the proposed guidance law. Figure 9b presents the change in the relative distance between the missile and the target and it can be observed that the miss distance is less than 0.5 m. By combining (a) and (b), the results reveal that the missile intercepts the maneuvering target successfully.

Curves of the angular rate and the range rate are presented in Figure 10. The results reveal the same conclusion as case 1.

Figure 11 presents the acceleration demands and the control requirements of the target and the missile. It can be observed that the lateral accelerations are maintained within a reasonable range, which can that ensure the missile intercepts the target. Furthermore, when the missile experiences external disturbances, the proposed robust differential game guidance strategy can successfully intercept maneuvering targets with external disturbances.

### 6.4. Effect of the Proposed Robust Optimal Differential Game Guidance Strategy with Other Methods

To confirm the advantage of the proposed robust optimal differential game guidance law, we offer a comparative experiment. In this experiment, the target selects the proposed robust optimal differential game guidance strategy (11), while the missile chooses the proposed differential game guidance strategy (11), i.e., the OGL in [4], and the conventional differential game guidance law (CDGGL) in [18], respectively. To further illustrate the advantages of the proposed differential game guidance strategy, the control effort *J* is defined as $J=\int_{0}^{\infty }u^{T}\left(\tau \right)u\left(\tau \right)d\tau .$ The comparison results for this engagement scenario are shown in Figure 12.

The engagement trajectories and the control effort of the missile are presented in Figure 12a. It can be observed that the OGL, the CDGGL, and the proposed robust optimal differential game guidance strategy both can successfully intercept the target, with a miss distance equal to 0.985, 1.218, and 0.0024 m, respectively. Furthermore, the curves of the missile control effort are presented in Figure 12b. It can be shown that the minimal control effort is our proposed robust optimal differential game guidance strategy, the second control effort is the CDGGL in [18], and the maximum control effort is the OGL in [4]. All control efforts are limited to the setting range (100 g). More importantly, it also can be observed that before the missile intercepts the target, the control effect we proposed is always at its minimum at different times. For example, when *t =* 4.8 s, the control effort of our method is minimal. Thus, the missile uses our method to intercept the target, which saves more energy. Furthermore, the OGL in [4] is larger than the proposed robust optimal differential game guidance strategy, which means that the missile may not successfully intercept the target when using the OGL in some acceleration-limited scenes. In general, our proposed guidance law is superior to the OGL in [4] and the CDGGL in [18]. To further illustrate the superiority of the proposed robust optimal differential game guidance strategy, the energy consumption and the simulation time are compared in 100 average experiments. The comparison results are shown in Table 1.

From Table 1, it is easily found that less control effort is needed in our proposed method compared with the OGL in [4] and the CDGGL in [18], which implies that our proposed guidance law can reduce unnecessary energy consumption. The missile can intercept the target with minimal control. Moreover, the computation time of our proposed method is shorter compared with the OGL in [4] and the CDGGL in [18], which means that our proposed method can make the confrontation strategy in the shortest time. In general, our proposed guidance law can intercept targets with minimal control effort and a fast response.

## 7. Conclusions

In this paper, a finite-time robust differential game law for the nonlinear two-player zero-sum game is proposed for unknown system dynamics with external disturbance. The robustness and optimality of the guidance strategy are proven under a time-varying cost function. In order to solve the fixed terminal time constraint, the HJI equation is solved by a critic NN with time-varying activation functions, and extra weight tuning terms are introduced to guarantee the stability of the closed-loop system and the minimization of the HJI approximation error. The NN identifier estimates the nonlinear system with an online tuning law, which can be subsequently utilized in the finite-time guidance strategy design. The proposed scheme yields an online guidance strategy design that enjoys great practical benefits. Finally, in the missile–target engagement experiments, the missile with the proposed differential game guidance law successfully intercepts the target in different forms of maneuvers and all the results verify the theoretical claims.

## Figures and Tables

**Figure 1 sensors-22-06650-f001:**
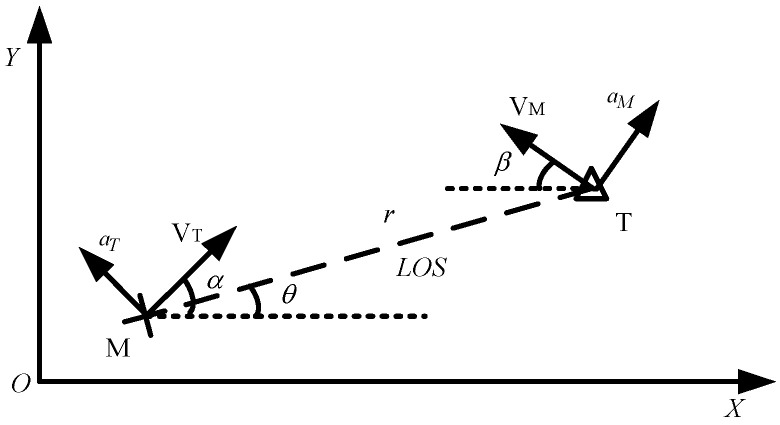
Missile–target engagement geometry.

**Figure 2 sensors-22-06650-f002:**
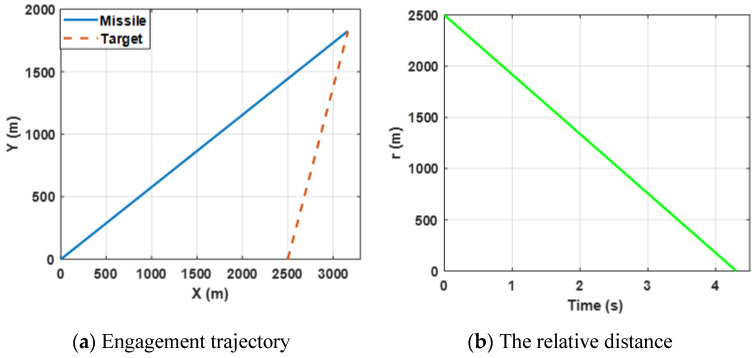
Engagement scenario for the proposed guidance strategy.

**Figure 3 sensors-22-06650-f003:**
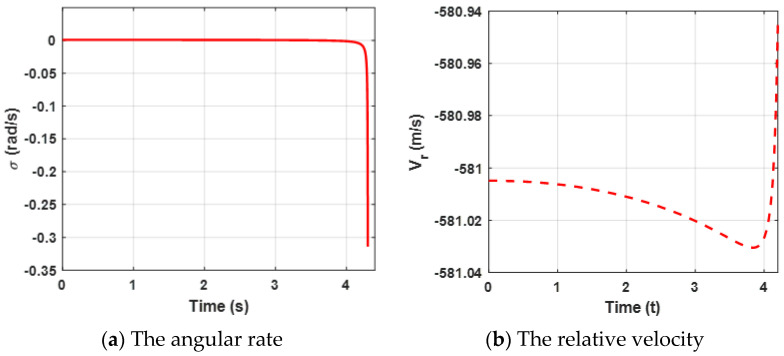
Curves of the angular rate and the relative velocity for the proposed guidance strategy.

**Figure 4 sensors-22-06650-f004:**
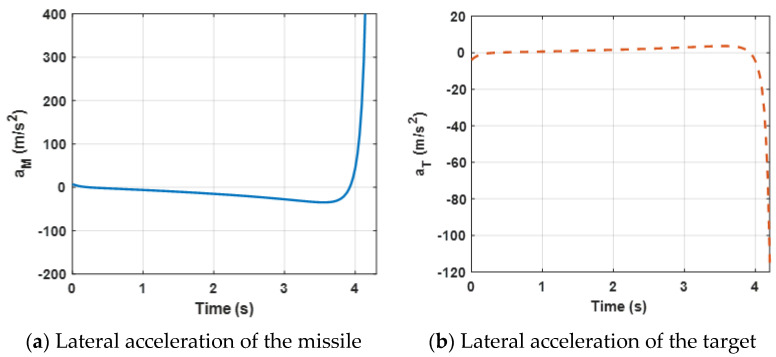
Acceleration demand of the missile and the target for the proposed guidance strategy.

**Figure 5 sensors-22-06650-f005:**
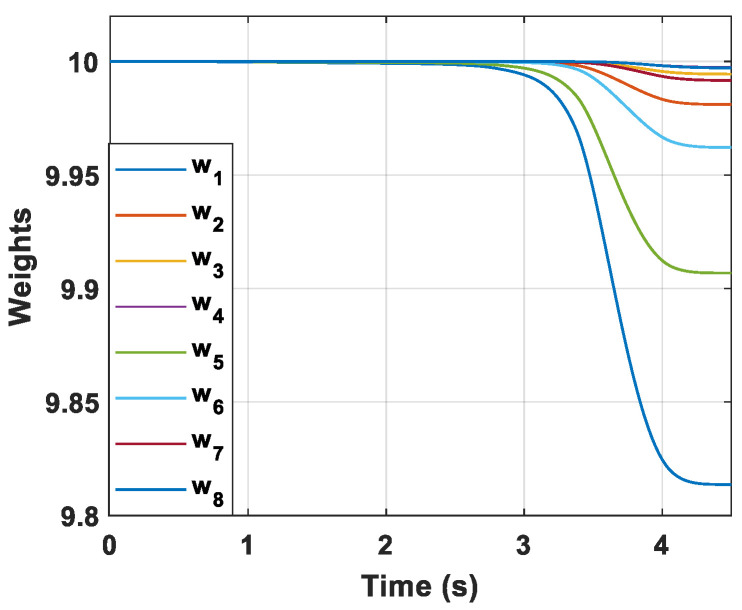
Convergence curves of critic NN weights.

**Figure 6 sensors-22-06650-f006:**
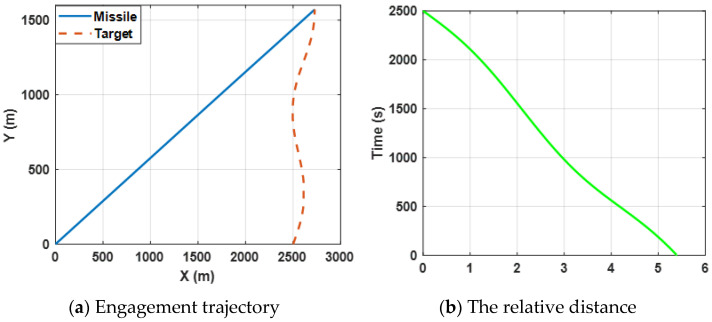
Engagement scenario for maneuvering target.

**Figure 7 sensors-22-06650-f007:**
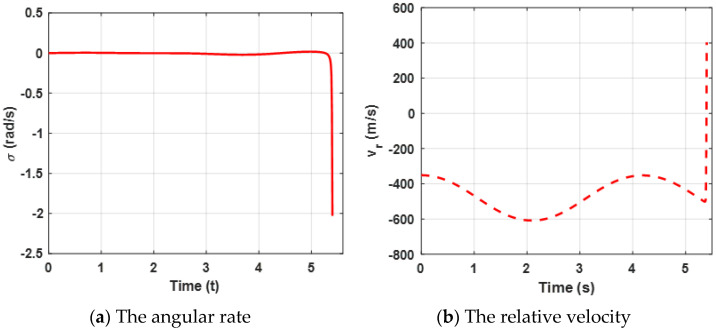
Curves of the angular rate and the relative velocity for a maneuvering target.

**Figure 8 sensors-22-06650-f008:**
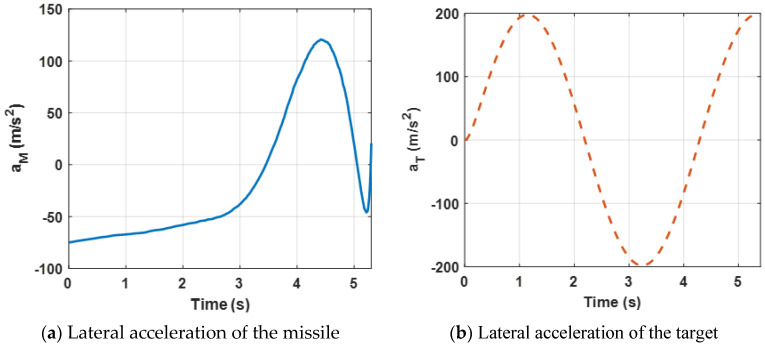
Acceleration demand of the missile and the target for a maneuvering target.

**Figure 9 sensors-22-06650-f009:**
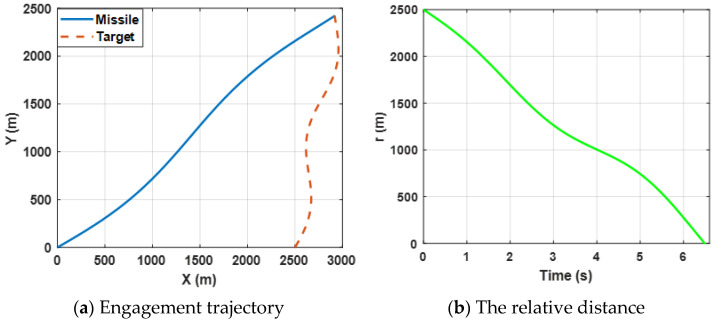
Engagement scenario for the target’s maneuverability.

**Figure 10 sensors-22-06650-f010:**
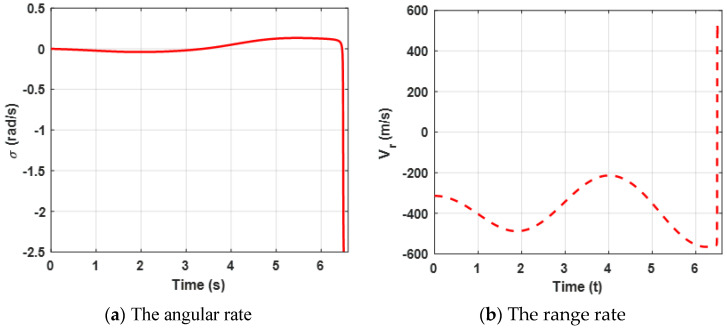
Curves of the angular rate and the relative velocity with unknown uncertainties.

**Figure 11 sensors-22-06650-f011:**
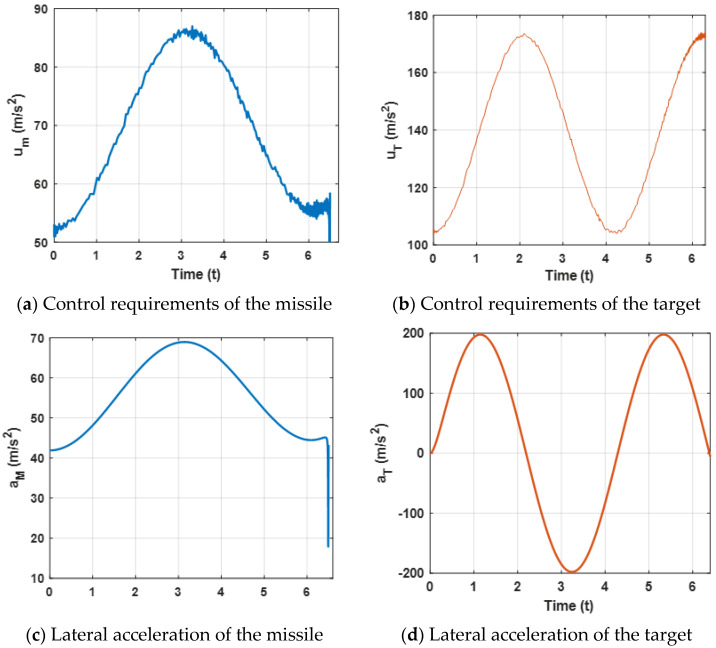
Acceleration demands of the missile and the target.

**Figure 12 sensors-22-06650-f012:**
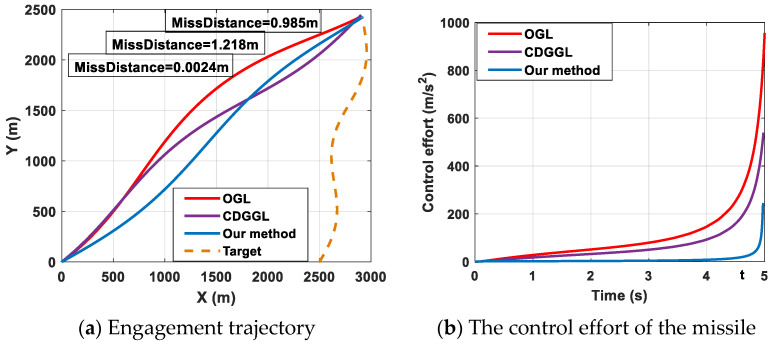
Engagement scenario and the control effort of the missile.

**Table 1 sensors-22-06650-t001:** Comparison results of OGL, CDGGL, and our method.

Method	Control Effort	Computation Time (t)	Number of Tests
OGL in [4]	980.56	1.647	100
CDGGL in [18]	570.64	1.083	100
Our method	218.845	0.451	100

## Data Availability

Not applicable.

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
