# Peer review of "A Nonlinear Finite-Time Robust Differential Game Guidance Law"

_sensors, 2022, doi:10.3390/s22176650_

Round 1
Reviewer 1 Report
This paper presents A Nonlinear Finite-Time Robust Differential Game Guidance Law. The work looks interesting. But, the paper needs to have huge revisions before it can be accepted as a Journal paper. Some comments to improve the manuscript.
1. The methodology should be presented clearly as a separated section.
2. It is difficult to figure out the technical novelty of this work throughout the manuscript. I recommend authors to clearly present the technical novelty in the introduction.
3. The theory part should be removed from the experimental section and experiment environments have to be briefly mentioned at the beginning of the experimental section.
Reviewer 2 Report
A robust differential game guidance law is proposed for the nonlinear zero- 10 sum system with unknown dynamics and external disturbances. For this paper, the following concerns are as:
(1) The detailed proof of (15) should be provided.
(2) The results seem be asymptotical ones ?
(3) The comparative analysis is wanted in simulation part.
Reviewer 3 Report
There are aspects that are worthy of publication, however, this paper requires a major attention and careful considerations. There are several things that need to be clarified.
1. The main contribution and originality should be explained in more detail, why is important?
2. More comparison of results against alternative approaches is needed for the benefit of the readers.
3. Authors should argue their choice of the performance evaluation indicators.
4. The Introduction could be updated with recent reviews dedicated to the references related to the topic addressed, particularly on fault detection approaches, influence of disturbances, modeling errors, various uncertainties in the real systems. A relevant recent review are: Online Reinforcement Learning Multiplayer Non-Zero Sum Games of Continuous-Time Markov Jump Linear Systems, Applied Mathematics and Computation; Asynchronous Fault Detection Observer for 2-D Markov Jump Systems, IEEE Transactions on Cybernetics; Exponential stability of nonlinear state-dependent delayed impulsive systems with applications, Nonlinear Analysis: Hybrid Systems; It is necessary to comment what would be changed in this case and make relation with the papers on this topic in Introduction section, and in that way, point out other contemporary approaches and possibilities. I believe this would further strengthen the introduction and lend support to the methodology applied in general.
5. The rationale on the choice of the particular set of parameters should be explained with more details. Have the authors experimented with other sets of values? What are the sensitivities of these parameters on the results?
6. English language should more polished and some typos corrected.
Round 2
Reviewer 1 Report
Please re-check the English.
Author Response
Thank you very much for your suggestion. We re-check our articles carefully, more details are marked in red in the revised manuscript
Reviewer 3 Report
The manuscript, in its present form, contains several weaknesses. Adequate revisions to the following points should be undertaken in order to justify recommendation for publication.
1. The advantages of the proposed method of this paper should be more highlighted.
2. Present a qualitative and quantitative comparative analysis of the proposed scheme with its conventional counterpart.
3. Authors should argue their choice of the performance evaluation indicators.
4. The Introduction could be updated with recent reviews dedicated to the references related to the topic addressed, particularly on fault detection approaches, influence of disturbances, modeling errors, various uncertainties in the real systems. A relevant recent review are: Online Reinforcement Learning Multiplayer Non-Zero Sum Games of Continuous-Time Markov Jump Linear Systems, Applied Mathematics and Computation; Exponential stability of nonlinear state-dependent delayed impulsive systems with applications, Nonlinear Analysis: Hybrid Systems; Value Iteration and Adaptive Optimal Output Regulation with Assured Convergence Rate, Control Engineering Practice. It is necessary to comment what would be changed in this case and make relation with the papers on this topic in Introduction section, and in that way, point out other contemporary approaches and possibilities. I believe this would further strengthen the introduction and lend support to the methodology applied in general.
5. The rationale on the choice of the particular set of parameters should be explained with more details. Have the authors experimented with other sets of values? What are the sensitivities of these parameters on the results?
Author Response
I find that the comments on round 1 and round 2 are almost identical. It makes me feel very confused. So I resubmitted the last response. If you find the resubmitted response is wrong, could you make the comments more specific? Thank you so much!
